# TSH Stimulation before PET/CT as Our Frenemy in Detecting Thyroid Cancer Metastases—Final Results of a Retrospective Analysis

**DOI:** 10.3390/cancers16193413

**Published:** 2024-10-08

**Authors:** Maciej Kołodziej, Marek Saracyn, Arkadiusz Lubas, Dorota Brodowska-Kania, Andrzej Mazurek, Mirosław Dziuk, Adam Daniel Durma, Stanisław Niemczyk, Grzegorz Kamiński

**Affiliations:** 1Department of Endocrinology and Isotope Therapy, Military Institute of Medicine—National Research Institute, 04-141 Warsaw, Poland; msaracyn@wim.mil.pl (M.S.); dbrodowska-kania@wim.mil.pl (D.B.-K.); adurma@wim.mil.pl (A.D.D.); gkaminski@wim.mil.pl (G.K.); 2Department of Internal Medicine, Nephrology and Dialysis, Military Institute of Medicine—National Research Institute, 04-141 Warsaw, Poland; alubas@wim.mil.pl (A.L.); sniemczyk@wim.mil.pl (S.N.); 3Department of Nuclear Medicine, Military Institute of Medicine—National Research Institute, 04-141 Warsaw, Poland; andrzej_mazurek@wim.mil.pl (A.M.); mdziuk@wim.mil.pl (M.D.)

**Keywords:** thyroid cancer, differentiated thyroid cancer, PET/CT, thyroid, rhTSH, radioiodine

## Abstract

**Simple Summary:**

Non-iodine avid differentiated thyroid cancer (DTC) may be a diagnostic problem, especially in patients with an indeterminate response to treatment (i.e., with thyroglobulin (Tg) concentration in the range of 1–10 ng/mL). PET/CT with [^18^F]FDG may be a useful technique for visualizing non-iodine avid DTC lesions. Presented study assessed the usefulness of using exogenous stimulation with recombinant human TSH (rhTSH) before PET/CT with [^18^F]FDG in detecting non-iodine avid foci of DTC in patients with elevated sTg and negative ^131^I WBS and the usefulness of Tg concentration in predicting a positive result of [^18^F]FDG PET/CT in this type of patients and to determine the optimal Tg cut-off point for obtaining a positive [^18^F]FDG PET/CT result. In a retrospective analysis including 73 patients with DTC suspected of having non-iodine avid recurrence or metastases there was no effect of rhTSH stimulation on the sensitivity and specificity of PET/CT, but a positive impact on the number of lesions visible in the examination was observed. Moreover, rhTSH stimulation allowed for visualization of non-iodine avid lesions with lower Tg concentration.

**Abstract:**

**Introduction:** Non-iodine avid metastases of differentiated thyroid cancer (DTC) can be found using PET/CT with a fluorine-18-labeled glucose analog ([^18^F]FDG). There are ongoing discussions on the appropriateness of using exogenous thyrotropin (TSH) stimulation before this examination. **Material and Methods:** In a retrospective study, 73 PET/CT scans with [^18^F]FDG performed after exogenous stimulation with recombinant human TSH (rhTSH) and without such stimulation were analyzed. All analyzed patients were suspected of having non-iodine-avid foci of DTC. **Results:** The stimulation with rhTSH before the PET/CT did not affect the percentage of positive results: 37.5% (18/48) with rhTSH and 40% (10/25) without rhTSH (*p* = 0.83). The analysis of the ROC curves established the cut-off thyroglobulin point for a positive PET/CT result separately for both subgroups. There was no statistically significant difference between obtaining a positive PET/CT result and the baseline thyroglobulin concentration (both stimulated and unstimulated). The exogenous stimulation of TSH prior to the PET/CT had no effect on the [^18^F]FDG uptake in the PET/CT lesions. **Conclusions:** PET/CT with [^18^F]FDG remains a useful method for the diagnosis of non-iodine-avid DTC lesions; in the presented group, a positive effect of rhTSH stimulation on the number of DTC foci visible in the PET/CT was found, but without affecting its effectiveness.

## 1. Introduction

In patients with differentiated thyroid cancer (DTC) after surgical treatment and adjuvant treatment with radioiodine (^131^I), which is the most useful methods recommended by European and American scientific societies to confirm remission and exclude recurrence and/or dissemination are monitoring the concentration of thyroglobulin (Tg) after TSH (thyroid stimulating hormone) stimulation (i.e., stimulated thyroglobulin—sTg), an ultrasound of the neck and diagnostic whole-body scintigraphy (WBS) with ^131^I [1]. Up to 30% of DTC patients with recurrence and/or metastasis may show a loss of iodine uptake by tumor cells [2,3]. Non-iodine-avid metastases are characterized by poorer differentiation, which results in their higher aggressiveness and greater growth dynamics. Clinical observations indicated a decrease in the percentage of 10-year survival in patients with non-iodine-avid metastases [2,4,5].

Suspicion of the presence of non-iodine-avid foci is an indication for PET/CT with the use of 2-[^18^F]Fluoro-2-deoxy-D-glucose ([^18^F]FDG). Such imaging shows quite high sensitivity and specificity in the search for non-iodine-avid foci of DTC: 68.4–100% and 66.7–98.5%, respectively [3,6,7,8,9,10,11]. Moreover, a clear correlation between an elevated sTg concentration and the presence of foci of high accumulation of [^18^F]FDG has been demonstrated in the literature [3,12,13]. There are well-established criteria based on robust data for determining which DTC patients need PET/CT with [^18^F]FDG scans [1,14], but there are ongoing discussions on the appropriateness of using TSH stimulation before this examination.

### Aim

In our center, some patients with DTC underwent PET/CT with [^18^F]FDG after TSH stimulation and some without stimulation. For this reason, the aim of this study was to assess the usefulness of using exogenous stimulation with recombinant human TSH (rhTSH) before PET/CT with [^18^F]FDG in detecting the non-iodine-avid foci of DTC in patients with elevated sTg and negative ^131^I WBS. The decision to perform a PET/CT examination not only on patients with an incomplete biochemical response to treatment (according to the American Thyroid Association (ATA) guidelines [1]) but also on patients with detectable Tg that was too low to determine an incomplete biochemical response to treatment—i.e., with an indeterminate response to treatment—was due to the need to identify patients with a higher risk of dedifferentiation of the disease with the presence of the DTC foci of higher aggressiveness.

The other objectives were to assess the usefulness of the Tg concentration in predicting a positive result of [^18^F]FDG PET/CT in this type of patient and to determine the optimal Tg cut-off point for obtaining a positive [^18^F]FDG PET/CT result in this group of patients.

## 2. Material and Methods

The Military Medical Chamber’s (Warsaw, Poland) Bioethics Committee approved this study. A retrospective analysis of 73 PET/CT examinations was performed. All of the PET/CT examinations took place between January 2018 and May 2024. This group included patients that were treated and diagnosed for differentiated thyroid cancer in the Department of Endocrinology and Isotope Therapy of the Military Institute of Medicine—National Research Institute (Warsaw, Poland). The characteristics of the analyzed groups are in Table 1.

A part of the results contained in the manuscript were previously published in 2021 in the journal *Nuclear Medicine Review* as a preliminary study [15].

The analyzed group included 61 patients (83.6%) with papillary thyroid cancer and 12 patients (16.4%) with follicular thyroid cancer. All patients underwent surgical treatment in the past and then ^131^I ablative treatment (in conditions of exogenous TSH stimulation). In the analyzed group, there were 54 patients (73.9%) who underwent more than one treatment with high ^131^I activity, including 31 patients (42.4%) who underwent ^131^I treatment two times, 14 patients (19.2%) who underwent ^131^I treatment three times, 3 patients (4.1%) treated with ^131^I four times, 3 patients (4.1%) treated with ^131^I five times, 2 patients (2.7%) treated with ^131^I six times and 1 patient (1.4%) treated with ^131^I seven times. The ^131^I activities used per single treatment ranged from 1.48 GBq to 7.4 GBq. The highest cumulative activity of 32.6 GBq was administered to the patient who was treated seven times.

For the retrospective analysis, we included PET/CT with [^18^F]FDG examinations in patients who met the following criteria:-Histopathologically confirmed DTC after a total thyroidectomy and ^131^I ablation treatment.-Non-stimulated and/or stimulated Tg concentration above 1.0 ng/mL (i.e., indeterminate or incomplete biochemical response to treatment according to the ATA guidelines [1]).-No foci of radioiodine uptake in diagnostic WBS.-No suspicious and/or pathological findings in the ultrasound examination of the neck.

Prior to the scintigraphy and sTg determination, all patients underwent stimulation with rhTSH administered intramuscularly twice at a dose of 0.9 mg, with an interval of 24 h.

Hormonal determinations (i.e., TSH and Tg) were made at the Department of Laboratory Diagnostics of the Military Institute of Medicine—National Research Institute using the electrochemiluminescence method (ECLIA).

An 8 MHz linear transducer of the Siemens Acuson X150 apparatus was used for the ultrasound examination.

WBS examinations were performed at the Department of Nuclear Medicine of the Military Institute of Medicine—National Research Institute using the parameters displayed in Table 2.

The PET/CT examinations were performed at the PET/CT Center—*Affidea* (Warsaw, Poland). A 64-row Discovery 710 hybrid scanner (GE Medical Systems) was used to perform the PET/CT. A PET/CT scan was performed 60 min after intravenous administration of the [^18^F]FDG. The administrated activity was 4 MBq/kg of the patient’s body weight. The radiotracer was obtained from a local cyclotron. The parameters used during the PET/CT with [^18^F]FDG are presented in Table 3.

The decision to perform the PET/CT with or without rhTSH stimulation depended on the availability of the [^18^F]FDG radiotracer and the availability of the time slot in the PET/CT Center’s schedule. If the [^18^F]FDG and the time slot were available during the patient’s hospitalization, the PET/CT was performed (using the rhTSH stimulation the patient received during hospitalization for a ^131^I whole-body scintigraphy and stimulated Tg determination) 24 h after the second injection of rhTSH. The PET/CT with rhTSH stimulation was performed 2 days before the WBS, prior to the 80 MBq ^131^I administration. The detailed sequence of the procedures performed is presented in Figure 1. In all patients included in the analysis, rhTSH was administered to perform the whole-body scintigraphy with ^131^I and the determination of stimulated Tg. Performing the PET/CT after rhTSH stimulation was a supplementary, additional procedure and no additional administration of rhTSH was used. None of the patients included in the analysis received rhTSH solely for performing the PET/CT.

If the [^18^F]FDG and the time slot were unavailable during the patient’s hospitalization, the patient, after the ^131^I whole body scintigraphy and sTg determination, was discharged, and the PET/CT examination was performed 14–28 days after, without rhTSH stimulation. Before this, the TSH was additionally confirmed to return to the suppression level (after the prior rhTSH stimulation, which the patient received during hospitalization for WBS with ^131^I and sTg determination). The detailed sequence of procedures performed is presented in Figure 2.

The analyzed group included 25 out of the total 73 (34.2%) examinations performed without rhTSH stimulation and 48 out of the total 73 (65.8%) examinations performed after rhTSH stimulation (rhTSH at a dose of 0.9 mg administered intramuscularly twice, with an interval of 24 h).

Each PET/CT image was evaluated by two independent nuclear medicine physicians with at least a Ph.D. and 10 years of clinical experience. DICOM format images were evaluated using the Volume Share 5—Advantage Workstation 4.6 multimodal workstation by General Electric Medical Systems. In the quantification of the PET/CT images, the maximum standardized uptake value (SUV_max_) calculated for lean body mass was used.

A positive PET/CT result was defined as one that showed at least one focus of increased FDG accumulation associated with DTC. Single or oligometastatic lesions visualized in the PET/CT suspected of being related to DTC were verified by fine-needle aspiration (FNA) with the determination of the Tg concentration in the needle wash after the FNA:-In the case of potentially resectable lesions, the lesions were operated on after the FNA verification (Figure 3a,b).-In the case of non-resectable lesions (due to a lack of the patient’s consent to repeated surgery, lack of technical possibilities of surgery or other reasons), the lesions were verified using FNA only, and after verification, the patients remained in the observation group or were referred to tyrosine kinase inhibitor (TKI) therapy (Figure 4a,b).

A Tg concentration in the needle wash after FNA that exceeded the Tg concentration in blood serum by at least two times was considered to confirm DTC. In rare cases of disseminated, widely spread disease with the presence of numerous foci of high, homogenous [^18^F]FDG accumulation in the PET/CT, and/or multiple non-resectable lesions, the patients qualified for TKI therapy after the verification of at least one lesion (Figure 5 and Figure 6a,b). The [^18^F]FDG tracer is not specific for DTC and may accumulate in inflammatory lesions and other malignancies; no other malignancies were found in the analyzed group of patients.

The obtained results are presented in the form of the mean with the standard deviation or the median with extreme values [15]. Nominal variables are presented as frequencies. The consistency of the distribution of variables with the normal distribution was checked using the Shapiro–Wilk test. The correlation analysis was performed using Pearson’s test for variables with a distribution close to normal; otherwise, Spearman’s test was used. Differences in the nominal variables between the groups were tested using the chi-square test, and quantitative variables were tested using Student’s *t*-test for unrelated variables. ROC analysis was performed to identify the cut-off points for the studied variables. The results of the tests performed were considered significant for two-sided *p* < 0.05. All statistical analyzes were performed using the *Statistica* v.12 package [15].

## 3. Results

Among the 73 PET/CT examinations with [^18^F]FDG, 25 examinations were performed without rhTSH stimulation and 48 were performed with rhTSH stimulation. In the subgroup of 25 PET/CT examinations without rhTSH stimulation, in 10 (10/25—40%), positive results of the PET/CT were obtained (Figure 7a,b). The other 15 studies in this subgroup (15/25—60%) were assessed as negative, and no foci of increased [^18^F]FDG accumulation associated with DTC were found.

In the subgroup of 48 PET/CT examinations after rhTSH stimulation, in 18 (18/48—37.5%), positive results were obtained (Figure 8a,b). The other 30 studies in this subgroup (30/48—62.5%) were assessed as negative, and no foci of increased [^18^F]FDG accumulation associated with DTC were found. The difference in the number of positive results in each subgroup was not statistically significant (*p* = 0.83).

In the subgroup of scans performed after the rhTSH stimulation, 122 foci of increased [^18^F]FDG accumulation associated with DTC were observed, while in the subgroup without rhTSH stimulation, 56 foci were observed (*p* < 0.05).

The mean SUV_max_ in the dominant lesion (i.e., the lesion with the highest accumulation of the radiotracer) visualized in the “unstimulated” subgroup was 7.63 ± 5.74, and in the “stimulated” subgroup, this was 9.24 ± 11.4 (the distribution did not meet the conditions of a normal distribution). The observed differences did not have statistical significance (*p* = 0.18). The median concentration of Tg is presented in Table 4. There was a statistically significant positive correlation between the natTg and sTg concentrations and the number of lesions in the PET/CT scans (r = 0.71, *p* < 0.05, and r = 0.70, *p* < 0.05, respectively).

Based on the analysis of the ROC curve, the optimal cut-off points for the concentrations of natTg and sTg for a positive PET/CT result were determined. With a sensitivity of 84.6%, a specificity of 79.1% and an accuracy of 81.2% for natTg, this value was 0.96 ng/mL (Figure 9). Moreover, with a sensitivity of 78.6%, a specificity of 82.2% and an accuracy of 80.8% for sTg, this value was 7.05 ng/mL (Figure 10). In addition, there was no statistically significant difference in the area under the ROC (AUC) curves between natTg and sTg (0.836 vs. 0.863, *p* = 0.45).

The analysis of the ROC curves in the subgroup showed that for a positive PET/CT result, the optimal cut-off points for the sTg concentration were as follows:-For the subgroup of examinations performed without rhTSH, it was 11.03 ng/mL (sensitivity 90%, specificity 93.3%, accuracy 92%).-For the subgroup of examinations performed after rhTSH stimulation, it was 6.3 ng/mL (sensitivity 77.8%, specificity 83.3%, accuracy 81.3%). However, this difference was not statistically significant (p = 0.17).

## 4. Discussion

PET/CT with [^18^F]FDG is recommended in patients with an elevated natTg and/or sTg concentration after surgery and ^131^I treatment, and no presence of abnormal and/or pathological lesions in imaging procedures (such as WBS and ultrasound of the neck). In such situations, this kind of examination is reimbursed in Poland by the National Health Fund (Polish: *Narodowy Fundusz Zdrowia*) [1]. Such indications for PET/CT with [^18^F]FDG can also be found in the American Thyroid Association (ATA) guidelines [14]. The aforementioned guidelines do not specify the level of natTg and/or sTg above which PET/CT is worth considering. However, the authors of these guidelines treat sTg levels above 10 ng/mL after prior radical treatment as an incomplete biochemical response to treatment. Regardless of the conditions (with or without rhTSH) in which the PET/CT was performed, we found a positive relationship between the concentration of thyroglobulin (both natTg and sTg) and the positive result of this imaging in the analyzed group. In the literature, one can find confirmation of such a relationship in the vast majority of available works in relation to sTg. Trybek et al. studied a group of 19 patients with DTC and showed a statistically significant accuracy of the sTg concentration in diagnosing the recurrence and/or metastasis of DTC [13]. In this group, all PET/CT examinations were performed after rhTSH stimulation, and positive examinations accounted for 31.6% (6/19), which is a lower result than ours, which was probably because our group was almost four times as numerous as the group presented by Trybek et al. A similar observation was made by Vural et al., who showed a positive relationship between the level of thyroglobulin (natTg and sTg) and a positive result of the PET/CT [8]. The analyzed group included 105 patients after ablative treatment due to DTC. All examinations were undertaken without rhTSH, and 71.4% (75/105) were positive. Of this group, a true positive result was confirmed in 69 patients (6 patients had a false positive result). In this group, the qualifying criterion for PET/CT was an sTg level above 10 ng/mL (in our study, above 1.0 ng/mL was used); hence, there was a significantly higher percentage of positive tests than in our analysis. In contrast, in 2007, Shammas et al. proved that the higher the thyroglobulin concentration, the higher the sensitivity of PET/CT (from 60% for thyroglobulin below 5 ng/mL to 62.5% for thyroglobulin ranging from 5 to 10 ng/mL and to 72% for thyroglobulin over 10 ng/mL) [7]. This analysis included 61 consecutive patients after a previous total thyroidectomy and ^131^I ablation therapy. The group included both patients with undetectable Tg (8 patients) and patients with elevated Tg (7 patients had elevated natTg, 46 patients had elevated sTg). Across the group, 49.2% (30/61) [^18^F]FDG PET/CT positivity rates were achieved (30/61), and the true positive rates increased with the Tg concentration (14% true positives at a Tg concentration below 5 ng/mL, 45% at a Tg concentration of 5–10 ng/mL and 62% with a Tg concentration above 10 ng/mL). In 2016, Stangierski et al. found that the probability of a positive PET/CT result increased with the increase in sTg concentration [16]. In this study, the authors retrospectively analyzed 69 [^18^F]FDG PET/CT scans performed without rhTSH stimulation. In 30 patients (43.5%), the foci of increased radiotracer accumulation were found, which is a result very similar to ours. It is worth noting that in the group presented by Stangierski et al., all patients had no foci of ^131^I accumulation in the post-therapeutic WBS performed after the administration of the ablative dose of ^131^I, while in the group presented by us, the qualifying criterion for the PET/CT with [^18^F]FDG was the negative result of WBS performed after the administration of the diagnostic activity of 80 MBq ^131^I.

In previous years, many authors attempted to determine the concentration of sTg at which the high probability of obtaining a positive PET/CT with [^18^F]FDG result would justify performing this examination. Such actions were undertaken mainly in order to limit using an expensive procedure with ionizing radiation to the group of patients in whom such an examination may be useful in making further therapeutic decisions. Na et al. analyzed 68 PET/CT scans in patients with papillary thyroid carcinoma with a negative WBS result and sTg elevated above 2 ng/mL [11]. Both the sTg determination and all PET/CT were performed in this group after levothyroxine (LT_4_) withdrawal (endogenous TSH stimulation), and the cut-off point for sTg by the authors was set at 20 ng/mL for a sensitivity of 85.7%. This is a higher value than that obtained by us, but in our group, none of the patients underwent sTg and PET/CT with [^18^F]FDG after the LT_4_ withdrawal. In our group, the rhTSH stimulation (instead of LT_4_ withdrawal) was used in all patients. Vural et al., in a study conducted on 105 patients, determined that the highest accuracy (sensitivity 74% and specificity 80%) was achieved at an sTg greater than 38.2 ng/mL and natTg greater than 1.9 ng/mL [8]. Additionally, they found that an sTg concentration greater than 38.2 ng/mL was independently related to FDG-avid DTC recurrence. The Polish group of Trybek et al. found that above an sTg concentration of 28.5 ng/mL, both the sensitivity and specificity of PET/CT were 100% [13]. In 2017, Chai et al., in a group of 240 patients with DTC who underwent PET/CT with [^18^F]FDG after at least 3 weeks of LT_4_ withdrawal (i.e., endogenous stimulation) and based on ROC curve analyses, determined the cut-off point for sTg at 49 ng/mL, with a sensitivity of 89.5% and a specificity of 90.9% [17]. Contrary to our group, the sTg concentration was determined without rhTSH. Meanwhile, in the previously cited study by Stangierski et al., this concentration was set at 32.9 ng/mL for sTg, but with a sensitivity below 60% and specificity slightly above 80% [16]. Based on the ROC analysis, our group assessed that the optimal concentration of thyroglobulin for obtaining a positive PET/CT result was 0.96 ng/mL for natTg (with sensitivity of 84.6% and specificity of 79.1%) and 7.05 ng/mL for sTg (with a sensitivity of 78.6% and a specificity of 82.2%). The concentrations of natTg and sTg presented by us were undoubtedly lower than those presented in other available literature studies and the works cited above, with a little lower but still acceptably high sensitivity. They were also lower than the values recommended by the Polish and American guidelines of DTC management [1,14]. This may have been due to the inclusion of our analyzed patients with a Tg (non-stimulated and/or stimulated) concentration above 1.0 ng/mL. Most of the studies cited above analyzed groups of patients with a Tg concentration above 10 ng/mL. According to the recommendations of the ATA, patients after ablative treatment due to DTC with an sTg concentration in the range of 1–10 ng/mL achieved an indeterminate response to the treatment [1]. Our study showed that PET/CT with [^18^F]FDG is also worth performing in this group of patients, especially in the case of negative results of WBS and a neck ultrasound.

It is worth noting that in our study, the differences in the ROC curves for natTg and sTg did not differ statistically significantly (*p* = 0.46), and therefore, the usefulness of natTg and sTg concentrations in predicting a positive PET/CT result in our group was comparable. Stimulation with rhTSH is an expensive procedure, where in Poland, it is available only during hospitalization. On the other hand, the natTg assay is a low-cost procedure that can be performed on an outpatient basis. Referring patients to a PET/CT with [^18^F]FDG based on the concentration of natTg (i.e., on a procedure that can be performed in outpatient clinics without the need for hospitalization) seems to potentially reduce the burden on the public healthcare system. A similar issue was of interest to the group of Prestwich et al. [18]. In a group of 58 rhTSH-PET/CT imaging examinations, the authors found that the sensitivity and specificity of the examinations with a natTg below 10 ng/mL were 44% and 73%, respectively, while with a natTg above 10 ng/mL, 87% and 100% were found, respectively. However, they also noted that in the analyzed group, there was no correlation between the natTg and the PET/CT result. This study presented by us, to our best knowledge, is probably the first in the available literature on the usefulness of the natTg concentration in predicting a positive PET/CT result with [^18^F]FDG in DTC patients.

Our retrospective analysis showed that [^18^F]FDG PET/CT performed after rhTSH stimulation was not superior to [^18^F]FDG PET/CT without rhTSH in detecting the foci of non-iodine avid recurrence and/or metastasis of DTC. Other authors also came to similar conclusions. Vera et al., in a group of 61 PET/CT examinations performed after rhTSH stimulation, obtained 41% (25/61) positive results [19]. In the study summary, the authors concluded that the sensitivity of PET/CT with [^18^F]FDG after rhTSH seemed to be low. Unlike our group, there were no PET/CT performed without rhTSH stimulation. In a prospective study, Leboulleux et al. assessed the impact of rhTSH administration on PET/CT in patients with DCT [20]. The analysis included 63 patients who underwent no-stimulation PET/CT, and then PET/CT after rhTSH stimulation. In this group, 49% (31/63) of patients had a positive examination result without rhTSH stimulation and 54% (34/63) of patients had a positive examination result after rhTSH stimulation (*p* = 0.42). A total of 35 patients (65%) had positive PET/CT results. Among them, in one patient (3%), lesions were visible only in the examination without rhTSH stimulation, and in four patients (11%), lesions were visible only in the examination after rhTSH stimulation. On the other hand, the authors reported that a total of 108 lesions were visualized—72 lesions (67%) in both examinations, 30 lesions (28%) only in the examination after rhTSH and 6 lesions (6%) only in the examination without rhTSH. The differences in the number of lesions identified in the study without rhTSH and with rhTSH (78 and 102 lesions, respectively) were statistically significant (*p* = 0.005). In both subgroups analyzed by us, a total of 178 lesions were visualized—56 lesions in the subgroup without rhTSH and 122 lesions in the subgroup with rhTSH. It should be noted that the subgroups in our study were not equal and this was not a head-to-head comparison. On the other hand, Kukulska et al. found that when rhTSH stimulation was used before PET/CT, the percentage of positive results of PET/CT in the group of patients with DTC increased [21]. The authors retrospectively analyzed 110 PET/CT examinations—25 examinations performed without stimulation and 85 examinations performed after stimulation (45 examinations with rhTSH and 39 examinations after LT_4_ withdrawal). In the non-stimulated subgroup, 28% (7/25) of the results were positive, and in the stimulated subgroup, 50% (42/85) were positive. In a meta-analysis of seven prospective studies (a total of 168 patients with DTC), Ma et al. showed that the probability of acquiring a positive PET/CT result undoubtedly increased when using either endogenous or exogenous TSH stimulation before this examination (OR= 2.45, 95% CI 1.23–4.9) [22]. Moreover, TSH stimulation (regardless of whether it was endogenous or exogenous) before PET/CT affected the number of visible foci of the high uptake of [^18^F]FDG (OR = 4.92, 95% CI 2.7–8.95). An increased uptake of [^18^F]FDG was expected after rhTSH stimulation due to the higher metabolic demand of the stimulated thyroid tissue. But, these results differ from those obtained in our study, where in the subgroup of examinations performed without rhTSH, the percentage of positive results of PET/CT was 40%, and in the subgroup of examinations performed after rhTSH, it was 37.5%. It should be noted, however, that in our retrospective analysis, PET/CT examinations without rhTSH and with rhTSH were performed on different groups of patients, whereas the studies cited above were head-to-head evaluations. Petrich et al. analyzed a group of 30 patients who underwent PET/CT without rhTSH stimulation and then PET/CT with rhTSH stimulation [23]. In this group, the authors obtained more than twice as many positive PET/CT results after using rhTSH (9 positive results without rhTSH vs. 19 positive results with rhTSH) [23]. Additionally, the rhTSH stimulation in this group before PET/CT statistically significantly increased the accumulation of the tracer (i.e., SUV_max_) in the visualized lesions and improved the tumor-to-background ratio (TBR). In 89.4% of patients (17/19) with a positive PET/CT result, the SUV_max_ increased after rhTSH stimulation, and in 94.7% (18/19), the TBR increased after stimulation. In addition, in 80% (24/30) of the patients that participated in the study, a decrease in the radiotracer uptake in soft tissues of the neck (background) was observed following the rhTSH injection. In our study, we also found a higher SUV_max_ in scans performed after the rhTSH stimulation, but we did not obtain statistical significance in the presented results (*p* = 0.18). Also, Almeida et al., in a head-to-head study of 15 patients with DTC who underwent PET/CT with [^18^F]FDG while using LT_4_ and 4 weeks after LT_4_ withdrawal (endogenous stimulation), found that in the post-stimulation study SUV_max_ of the visible lesions was higher, but the difference was not statistically significant—not for nodal (*p* = 0.43) nor distant metastases (*p* = 0.75) [24].

We performed an evaluation of our ROC curves separately for “unstimulated” and “stimulated” subgroups, where we found that for a positive PET/CT result, the cut-off point for sTg in the “unstimulated” subgroup (i.e., without rhTSH) was 11.03 ng/mL, while in the “stimulated” subgroup (i.e., with rhTSH), it was 6.3 ng/mL. Surprisingly, this clear difference was not statistically significant (*p* = 0.17). This was mainly due to the small sizes of the analyzed subgroups (10 positive PET/CT results were obtained in the subgroup without rhTSH and 18 in the subgroup with rhTSH). Despite the lack of statistical significance of the observed difference, it could be concluded that PET/CT after rhTSH is particularly recommended in patients with slightly elevated sTg levels (in our study, this was in the range of 6.3–11.03 ng/mL). According to the guidelines of ATA patients with these sTg concentrations, after a complete thyroid resection and ^131^I ablative therapy, they have an indeterminate response to treatment, which may be a problem in determining further diagnostic and therapeutic procedures [14]. However, in a patient whose sTg concentration exceeded 11.03 ng/mL, the use of rhTSH before PET/CT seemed to be not necessary. In the available literature, no large studies that referred to the possibility of using rhTSH stimulation before PET/CT exclusively in patients with indeterminate response (according to ATA guidelines) were found. In our opinion, this is an issue that requires future prospective research.

The main advantages of our work were the presentation of a statistically significant relationship between the Tg concentration and a positive PET/CT result, the demonstration that the Tg concentration with TSH suppression had a comparable value in predicting a positive PET/CT result as the Tg concentration after rhTSH stimulation, and the determination of cut-off points for Tg for both PET/CTs done with and without rhTSH stimulation. The results obtained in our work may be translated into everyday clinical practice and may be a contribution to further scientific research.

The limitation of our work was its retrospective nature and the lack of a head-to-head evaluation of the obtained PET/CT results with [^18^F]FDG. Another limitation of the presented study was the fact that the decision to perform a PET/CT examination with [^18^F]FDG during hospitalization, and thus, in the conditions of rhTSH stimulation (and thus, assigning the patient to one of two subgroups), was dependent only on the availability of the radiotracer [^18^F]FDG and a free time slot at the PET/CT Center’s schedule, without taking into account the data from the patient’s medical history (such as the Tg concentration).

## 5. Conclusions

In our study, we showed that PET/CT with [^18^F]FDG is a beneficial diagnostic tool for localization recurrence and/or the metastasis of differentiated thyroid cancer without iodine avidity. Although the administration of rhTSH before this imaging did not improve its effectiveness in diagnosing the abovementioned lesions compared with the one performed without rhTSH, it allowed for the visualization of a larger number of lesions with a lower concentration of stimulated Tg.

Accumulation of the tracer visible in PET/CT, which may correspond to recurrence and/or metastases, correlated positively with the unstimulated and stimulated thyroglobulin concentration, and the values of both in predicting a positive PET/CT result were comparable.

The optimal cut-off point for a positive PET/CT result in the presented group for unstimulated thyroglobulin concentration was 0.96 ng/mL, and for stimulated thyroglobulin, it was 7.05 ng/mL.

The presented results encourage further research, especially a head-to-head comparison, in which each patient would undergo a PET/CT scan before and after rhTSH administration.

Key Points:

Question: Does rhTSH stimulation prior to [^18^F]FDG PET/CT in a patient with non-iodine-avid differentiated thyroid carcinoma (DTC) improve the sensitivity and specificity of this examination?

Pertinent findings: In a retrospective analysis that compared [^18^F]FDG PET/CT examinations performed with and without rhTSH stimulation, no significant effect of rhTSH stimulation on the sensitivity and specificity of this examination was found; however, the cut-off point for stimulated thyroglobulin in the subgroup of examinations performed after rhTSH stimulation was lower, whereas the number of visualized lesions was higher.

Implications for patient care: The use of rhTSH stimulation prior to [^18^F]FDG PET/CT examination may be limited to patients with non-iodine-avid DTC in whom the serum concentration of stimulated thyroglobulin is moderately elevated (i.e., in the range of 6.3–11.03 ng/mL).

## Figures and Tables

**Figure 1 cancers-16-03413-f001:**
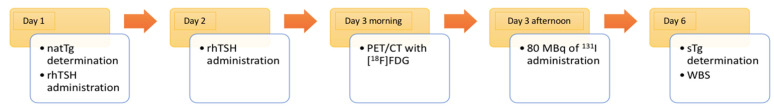
The detailed sequence of procedures performed in patients in whom PET/CT had been performed after rhTSH stimulation. The patient was admitted to the ward on day 1 and discharged on day 6. PET/CT was performed during hospitalization. natTg—native thyroglobulin, sTg—stimulated thyroglobulin, WBS—whole body scintigraphy.

**Figure 2 cancers-16-03413-f002:**
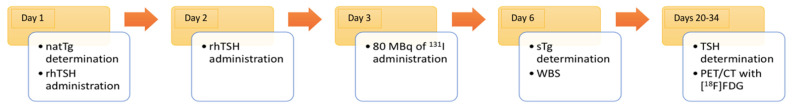
The detailed sequence of procedures performed in patients in whom PET/CT had been performed without rhTSH stimulation. The patient was admitted to the ward on day 1 and discharged on day 6. PET/CT was performed as an outpatient procedure. natTg—native thyroglobulin, sTg—stimulated thyroglobulin, WBS—whole-body scintigraphy.

**Figure 3 cancers-16-03413-f003:**
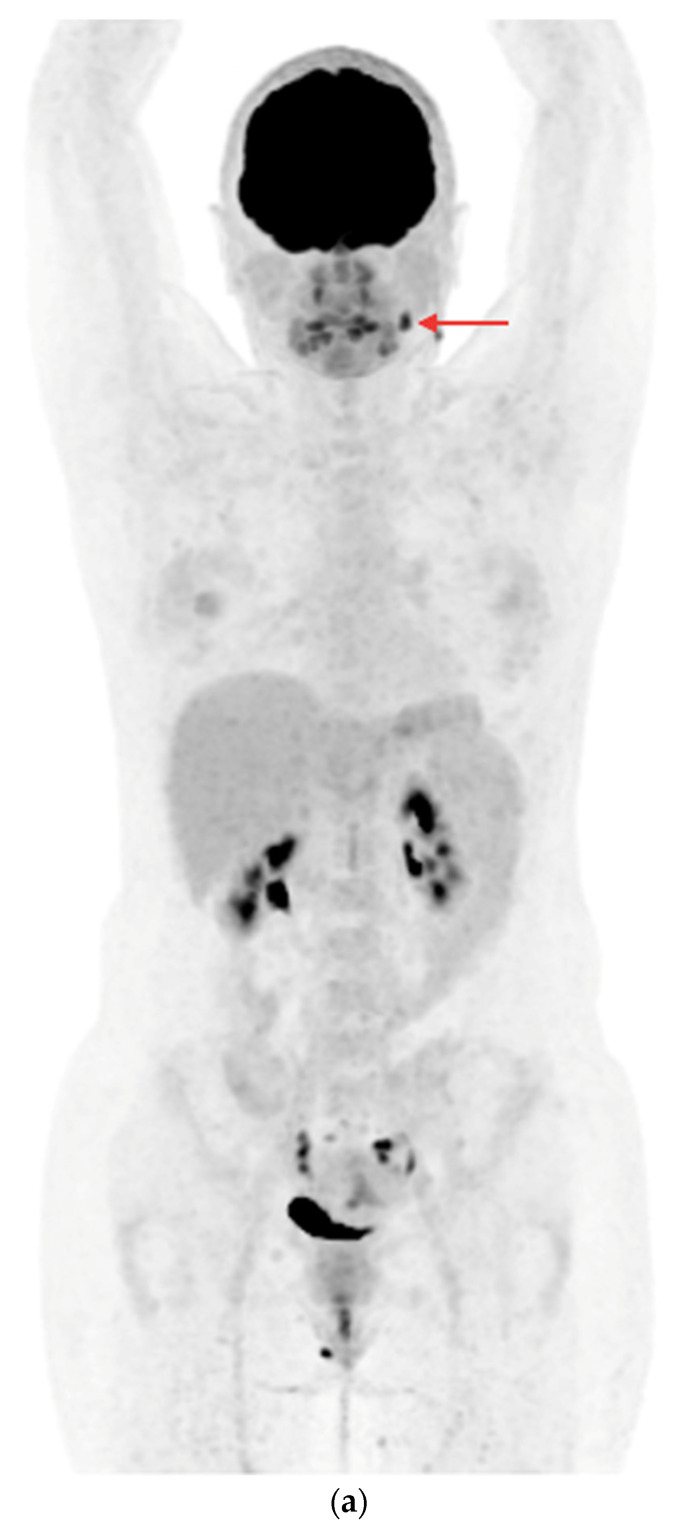
PET/CT with [^18^F]FDG (**a**) Maximum-intensity projection (MIP). (**b**) Axial fusion projection. These show an example of the accumulation of the [^18^F]FDG in the left cervical lymph node (red arrow). The lesion was verified as being metastasized DTC using FNA and resected. The preoperative sTg concentration was 5.55 ng/mL. In the postoperative follow-up, the sTg concentration decreased to being undetectable (<0.04 ng/mL).

**Figure 4 cancers-16-03413-f004:**
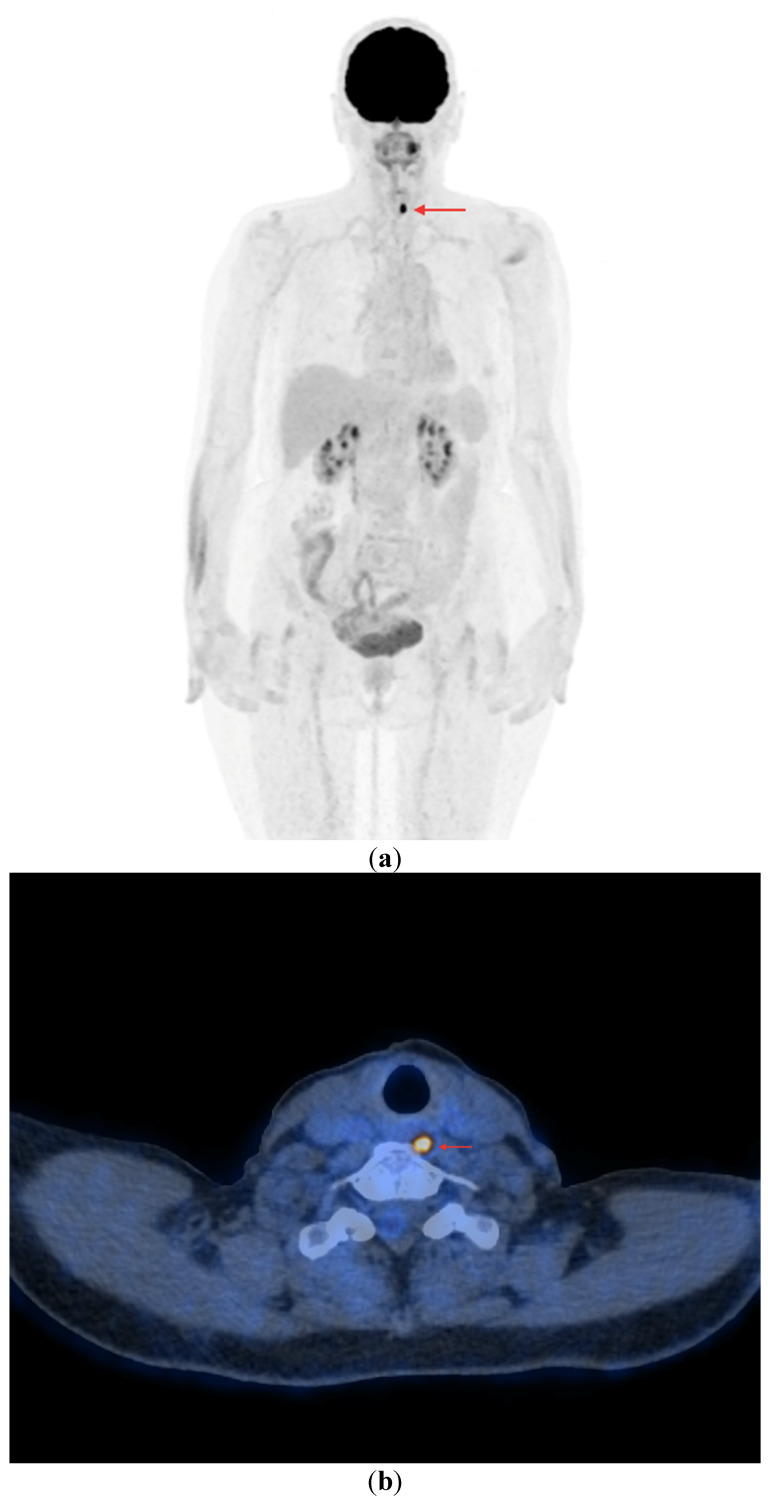
PET/CT with [^18^F]FDG (**a**) Maximum-intensity projection (MIP). (**b**) Axial fusion projection. These show an example of accumulation of the [^18^F]FDG in the left retroesophageal lymph node (red arrow). The lesion was verified as being metastasized DTC using FNA. Due to comorbidities and the patient’s lack of consent, resection of the lesion was abandoned; the patient was left in the observation group, with a stable Tg concentration in follow-up determinations (maximum natTg concentration 3.91 ng/mL, maximum sTg concentration 19.62 ng/mL).

**Figure 5 cancers-16-03413-f005:**
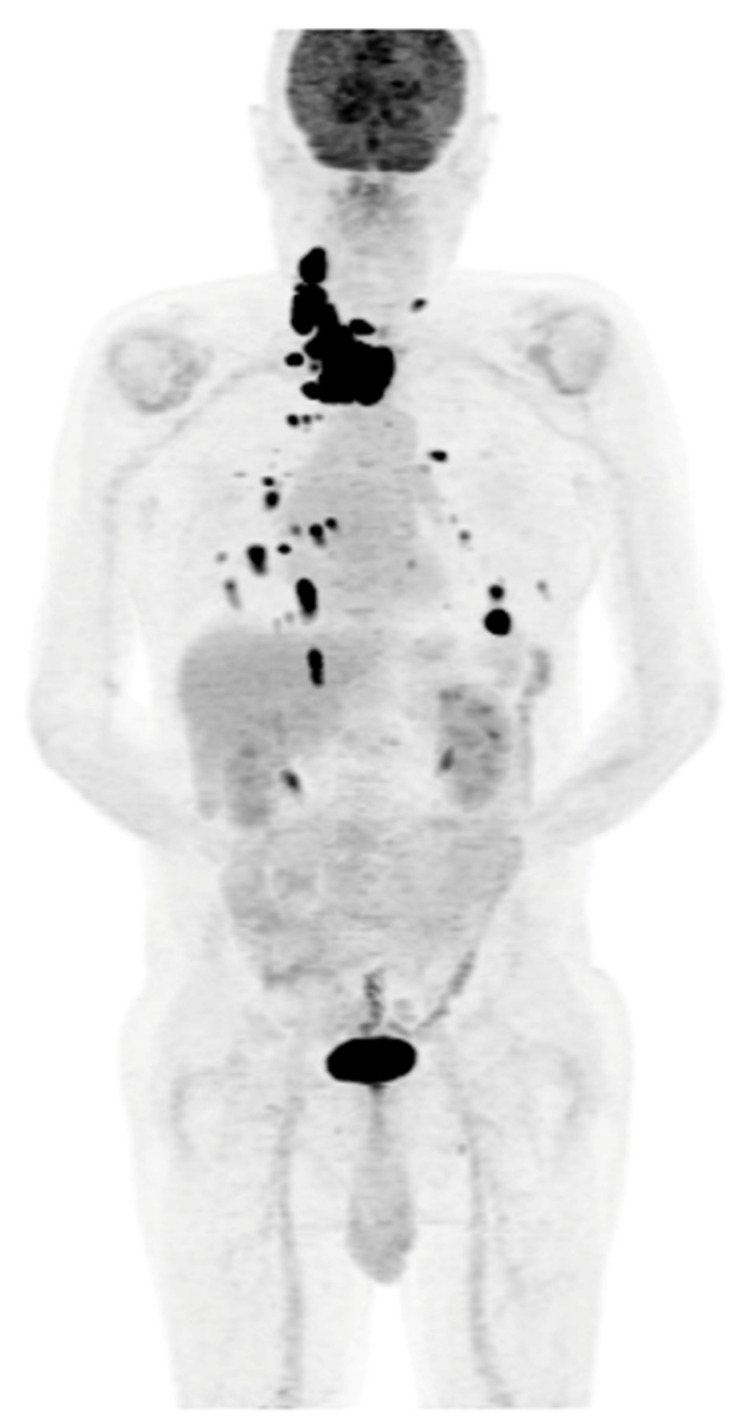
Maximum-intensity projection (MIP) PET/CT with [^18^F]FDG. Multiple foci of increased [^18^F]FDG accumulation are visible in the neck (unresectable local recurrence), with the presence of metastatic lesions in the lymph nodes and in both lungs. Patient qualified for TKI (sorafenib) therapy with stable disease since 2022.

**Figure 6 cancers-16-03413-f006:**
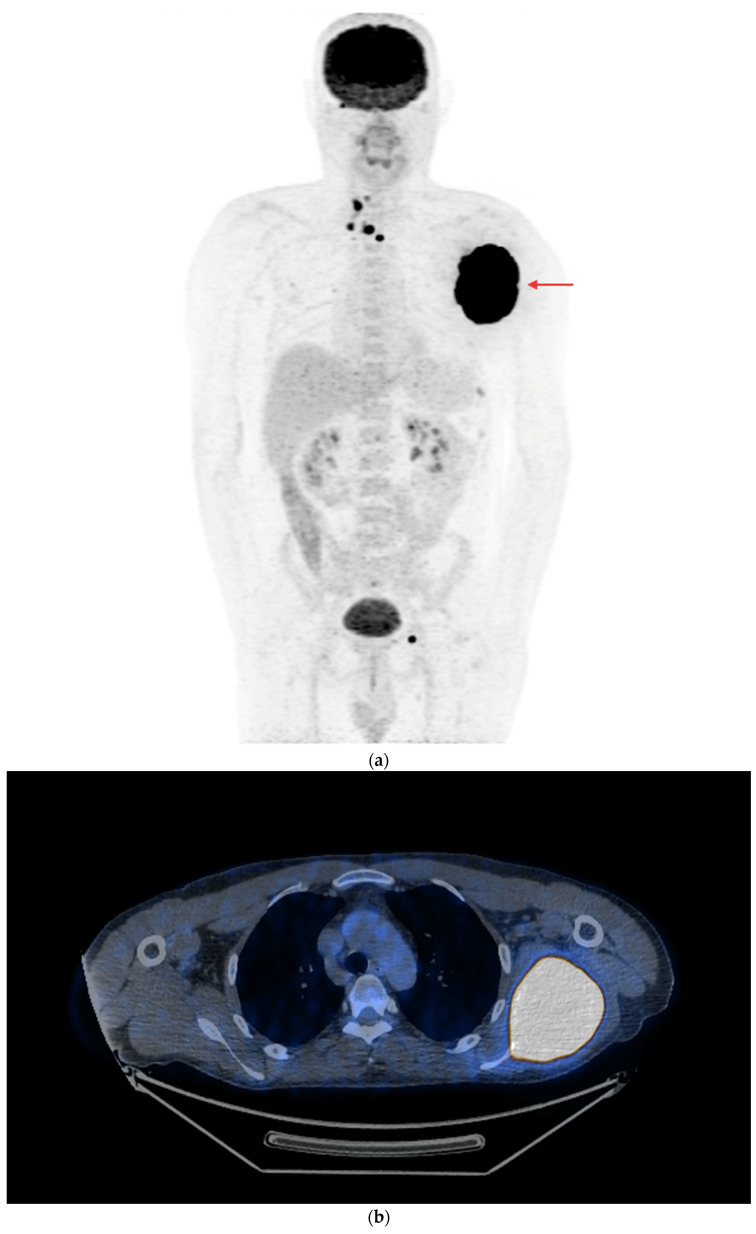
PET/CT with [^18^F]FDG (**a**) Maximum-intensity projection (MIP). (**b**) Axial fusion projection. Multiple metastatic lesions are visible in the mediastinal lymph nodes, with a small metastasis to the left pelvic bone and an extensive tumor in the left scapula (red arrow) is visible with high [^18^F]FDG accumulation (the lesion was confirmed using a core needle biopsy as metastasized DTC). The patient was disqualified from surgical treatment and qualified for TKI therapy (initially sorafenib, then cabozantinib due to progression). The patient died in December 2023 (i.e., 51 months after the DTC diagnosis and 37 months after the confirmation of non-iodine-avidity disease).

**Figure 7 cancers-16-03413-f007:**
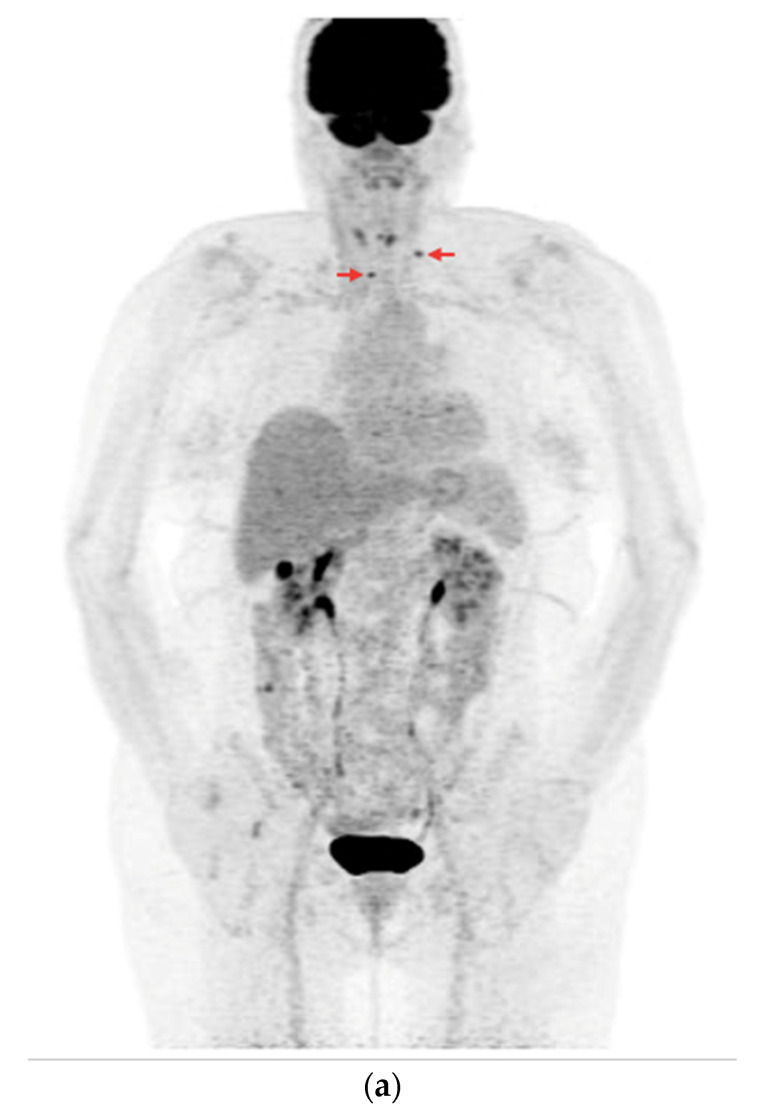
PET/CT with [^18^F]FDG without rhTSH stimulation (**a**) Maximum-intensity projection (MIP). An example of accumulation of the [^18^F]FDG in the lymph nodes: left cervical and right mediastinal (red arrows). (**b**) Axial fusion projection. An example of accumulation of the [^18^F]FDG in the right mediastinal lymph node (red arrow).

**Figure 8 cancers-16-03413-f008:**
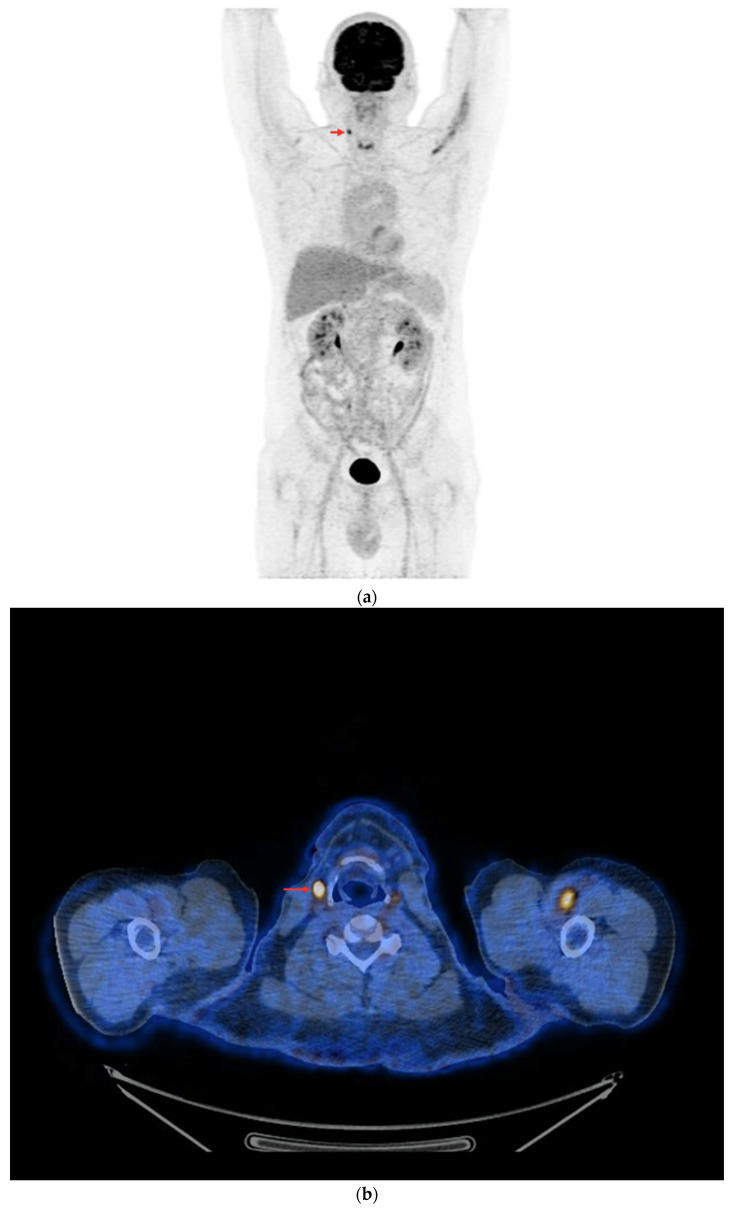
PET/CT with [^18^F]FDG with rhTSH stimulation (**a**) Maximum-intensity projection (MIP). (**b**) Axial fusion projection. These show an example of accumulation of the [^18^F]FDG in the right cervical lymph node (red arrow).

**Figure 9 cancers-16-03413-f009:**
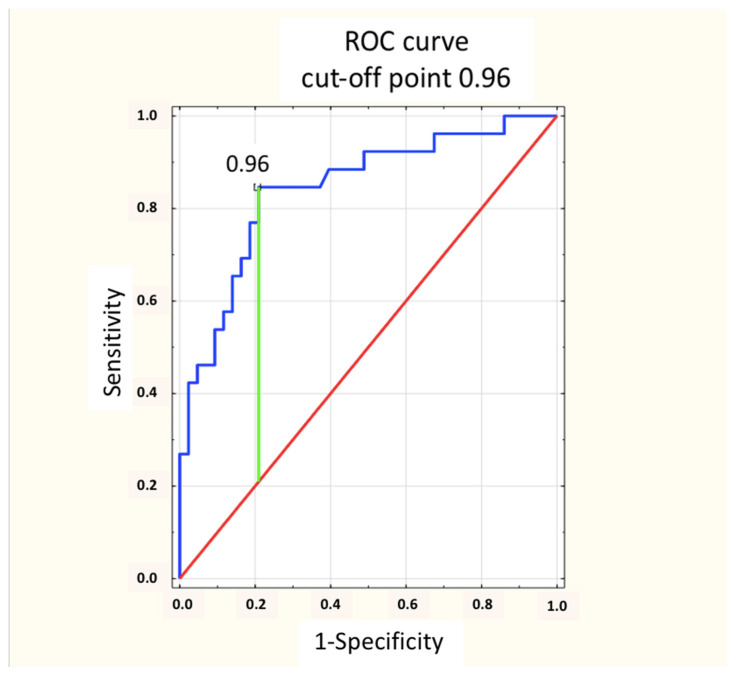
The concentration of natTg ROC curve—the optimal cut-off point for the concentration of natTg for a positive PET/CT result was 0.96 ng/mL (sensitivity of 84.6% and specificity of 79.1%). Blue line: ROC curve; Red line: Chance level; Green line: Maximum value of Youden’s index for the ROC curve.

**Figure 10 cancers-16-03413-f010:**
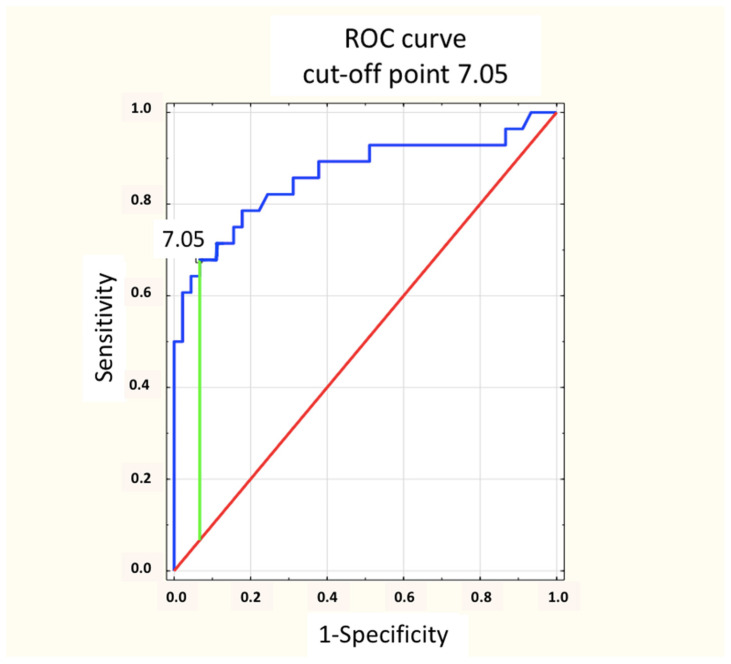
The concentration of sTg ROC curve showing the optimal cut-off point for a positive PET/CT result was 7.05 ng/mL (sensitivity of 78.6%, specificity of 82.2%). Blue line: ROC curve; Red line: Chance level; Green line: Maximum value of Youden’s index for the ROC curve.

**Table 1 cancers-16-03413-t001:** Characteristics of the analyzed groups.

Analyzed Feature	PET/CT w/o rhTSH (*n* = 25)	PET/CT with rhTSH (*n* = 48)	*p*
Mean age (years)	48	50	0.54
Sex	F = 22, M = 3	F = 33, M = 15	n/a
Number of patients with repeated therapy	16	38	0.53
Mean number of treatments	2.3	2.3	1.0
Mean activity per therapy (GBq)	4.08	4.26	0.75
Mean total activity (GBq)	9.46	9.93	0.58
Histology type:			
- Papillary	20	41	n/a
- Follicular	5	7	n/a
Tumor size (acc. TNM):			
- T1a	2	4	n/a
- T1b	6	7	n/a
- T1m	6	13	n/a
- T2	2	5	n/a
- T3 and T4	8	17	n/a
- Tx	1	2	n/a
Lymph nodes (acc. TNM):			
- N0	7	20	n/a
- N1	15	21	n/a
- Nx	3	7	n/a
Metastases (acc. TNM):			
- M0	19	43	n/a
- M1	6	5	n/a

TNM—Classification of Malignant Tumors TNM 8th edition, rhTSH—recombinant human TSH, w/o—without, acc.—according, n/a—not applicable.

**Table 2 cancers-16-03413-t002:** Selected parameters used during WBS in presented group.

Feature	Value
Device	NM/CT 870DR (GE Medical Systems, Waukesha, WI, USA)
Detector configuration	H-mode
Energy window	364 keV ± 10%
Collimator	High-energy general purpose
Body contour	On
Scan mode	Continuous
Exposure time per pixel	320 s
Speed	7 cm/min
Matrix	256 × 1024
Acquisition zoom	1

keV—kiloelectron volt, s—second.

**Table 3 cancers-16-03413-t003:** Selected parameters used during the PET/CT in presented group.

Tomographic Part	Emission Part
Voltage: 140 kV	Subsets: 18
Current: 40–100 mA	Iterations: 3
Noise index: 22	Matrix: 256 × 256
Thickness of the tomographic layer: 1.25 mm	Time-of-flight: on
Time lamp rotation: 0.8 s	
Pitch: 0.984:1	

kV—kilovolt, mA—milliampere, s—second.

**Table 4 cancers-16-03413-t004:** The median concentrations of thyroglobulin in presented subgroups.

	Subgroup PET/CT without rhTSH	Subgroup PET/CT with rhTSH
Median natTg [ng/mL] (min–max)	1.0 (0.04–500)	0.61 (0.04–6779)
Median sTg [ng/mL] (min–max)	7.18 (1.45–28,287)	4.95 (0.56–7079)

natTg—thyroglobulin concentration with TSH suppression, sTg—thyroglobulin concentration after rhTSH stimulation, min—minimal, max—maximal, rhTSH—recombinant human TSH.

## Data Availability

Data other than that published in the manuscript are partially unavailable due to privacy or ethical restrictions.

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
