# Peer review of "TSH Stimulation before PET/CT as Our Frenemy in Detecting Thyroid Cancer Metastases—Final Results of a Retrospective Analysis"

_cancers, 2024, doi:10.3390/cancers16193413_

Round 1
Reviewer 1 Report
Comments and Suggestions for Authors
The selection criteria of the patients schould be clarified. What was the indication of the investigations with such low Tg levels, without morphological suspect lesions? What was the folow up results of the FDG PET/CT positive patients? The criteria of positive PET scan is not clear (any FDG positive lesion?). The number of detected lesions on FDG PET/CT were much higher with rh TSH than without (122 vs 56 mentioned only in discussion). This schould be further analysed. The detected FDG positive lesions were histologically verified?
Author Response
The selection criteria of the patients schould be clarified. What was the indication of the investigations with such low Tg levels, without morphological suspect lesions?
According to the ATA response criteria for treatment of patients with DTC, an unstimulated Tg above 1 ng/mL or a stimulated Tg above 10 ng/mL or the presence of persistent disease in imaging techniques means of an incomplete biochemical or structural response. From a clinical point of view, the biggest challenge are patients with the so-called indeterminate response to treatment (i.e., those with stimulated Tg concentration is in the range of 1-10 ng/mL or unstimulated Tg is detectable but lower than 1ng/mL, and in imaging techniques (WBS and US) no presence of persistent disease is detected). Performing a PET/CT in patients with detectable Tg, but too low to determine an incomplete biochemical response to treatment, i.e., with an indeterminant response to treatment (especially in the absence of radioiodine uptake in scintigraphy), allows the identification of patients with a higher risk of dedifferentiation of the disease, and therefore with a disease with a higher aggressiveness.
An explanation has been added to the manuscript.
The criteria of positive PET scan is not clear (any FDG positive lesion?). What was the folow up results of the FDG PET/CT positive patients?
A positive PET/CT scan is one that shows at least one focus of increased FDG accumulation associated with DTC.
Of course, the FDG tracer is not specific for DTC and may accumulate in inflammatory lesions and other cancers. No other malignancies were found in the analyzed group of patients. An explanation has been included in the manuscript.
Changes suspected of being related to DTC:
- in the case of single or multiple lesions, but resectable - the lesions were operated on (with histopathological confirmation by DCT) or first verified by FNA (with the determination of Tg concentration in the needle wash after FNA), and then operated on;
- in the case of limited but non-resectable lesions (lack of patient's consent to repeated surgery, lack of technical possibilities of surgery or other), the lesions were verified in FNA (with the determination of Tg concentration in the needle wash after FNA) and after confirming that these changes were related to DTC patients remained in the observation group or were referred to TKI therapy;
- in the case of disseminated lesions, the presence of numerous metastases in PET/CT, unquestionable lesions, and numerous non-resectable lesions, patients were qualified for TKI therapy.
An appropriate explanation along with sample PET/CT images has been added to the manuscript.
The number of detected lesions on FDG PET/CT were much higher with rh TSH than without (122 vs 56 mentioned only in discussion).
Thank you for this remark. The appropriate paragraph has been added to the manuscript in the Results section and the Conclusions have been modified.
Reviewer 2 Report
Comments and Suggestions for Authors
This study assessed the usefulness of using exogenous stimulation with recombinant human TSH (rhTSH) before PET/CT with [18F] FDG in detecting non-iodine avid foci of DTC in patients with elevated sTg and negative 131I WBS and the usefulness of Tg concentration in predicting a positive result of [18F] FDG PET/CT in this type of patients and to determine the optimal Tg cut-off point for obtaining a positive [18F] FDG PET/CT result in this group of patients. This topic is relatively useful and novel. The result is benefit for clinical decision. But there are some issues the authors should be work on:
(1) The grammar and expression in the most part of the manuscript is confusing, making it difficult to understand the article.
(2) By compare the value of PET/CT with [18F] FDG in the two group, the author used the mean SUVmax in the dominant lesion. But the two group is not the direct (head-to-head) comparison like the author says. And there is no gold standard like biopsy to confirm the recurrence or metastasis. So, there may be false negatives and false positives for PET/CT with [18F] FDG. So, this result cannot conclude that PET/CT with [18F] FDG performed after rhTSH stimulation does not improve its effectiveness in diagnosing the above-mentioned changes compared to the one performed without such stimulation. I suggest the authors revise the statistical results.
Comments on the Quality of English LanguageThe grammar and expression in the most part of the manuscript is confusing, making it difficult to understand the article.
Author Response
(1) The grammar and expression in the most part of the manuscript is confusing, making it difficult to understand the article.
The English grammar and language have been checked again - appropriate corrections have been made to the manuscript.
(2) By compare the value of PET/CT with [18F] FDG in the two group, the author used the mean SUVmax in the dominant lesion. But the two group is not the direct (head-to-head) comparison like the author says. And there is no gold standard like biopsy to confirm the recurrence or metastasis. So, there may be false negatives and false positives for PET/CT with [18F] FDG. So, this result cannot conclude that PET/CT with [18F] FDG performed after rhTSH stimulation does not improve its effectiveness in diagnosing the above-mentioned changes compared to the one performed without such stimulation. I suggest the authors revise the statistical results.
Indeed, the two compared groups differed from each other, it was not a head-to-head study (then each patient would have to undergo a PET/CT examination before and after rhTSH administration). The physiology of rhTSH action could suggest that in the post-rhTSH study, FDG uptake in DTC-related lesions will be higher (this was also the case in our observation, but without statistical significance) - this is mentioned in our manuscript, but it is not the main observation from our study.
The second reviewer pointed out that despite the comparable percentage of positive PET/CT results in both subgroups, the rhTSH subgroup showed significantly more DTC-related lesions than the non-rhTSH group (122 and 56, respectively) - these data were included in Results, and the study conclusions were modified in the manuscript.
Of course, the FDG tracer is not specific for DTC and may accumulate in inflammatory lesions and other cancers. No other malignancies were found in the analyzed group of patients. An explanation has been included in the manuscript.
Changes suspected of being related to DTC:
- in the case of single or multiple lesions, but resectable - the lesions were operated on (with histopathological confirmation by DCT) or first verified by FNA (with the determination of Tg concentration in the needle wash after FNA), and then operated on;
- in the case of limited but non-resectable lesions (lack of patient's consent to repeated surgery, lack of technical possibilities of surgery or other), the lesions were verified in FNA (with the determination of Tg concentration in the needle wash after FNA) and after confirming that these changes were related to DTC patients remained in the observation group or were referred to TKI therapy;
- in the case of disseminated lesions, the presence of numerous metastases in PET/CT, unquestionable lesions, and numerous non-resectable lesions, patients were qualified for TKI therapy.
An appropriate explanation along with sample PET/CT images has been added to the manuscript.
Yes, each negative PET/CT result in a patient with elevated Tg concentration and negative WBS can be assessed as a false negative (since we detect Tg, we should have cells producing it somewhere) - such patients remain under close observation. A negative PET/CT result allows us to exclude the presence of more aggressive, dedifferentiated disease lesions in these patients (or at least those consisting of more than 10^8 cancer cells - the PET/CT resolution allows for imaging cancer lesions usually consisting of at least 10^8 cells).
Round 2
Reviewer 1 Report
Comments and Suggestions for Authors
I accept ths corrections
Reviewer 2 Report
Comments and Suggestions for Authors
Thanks for the response. The manuscript can be accepted in present form.